# Artificial intelligence applied to bed regulation in Rio Grande do Norte: Data analysis and application of machine learning on the "RegulaRN Leitos Gerais" platform

Tiago de Oliveira Barreto[1]*, Fernando Lucas de Oliveira Farias[1], Nicolas Vinícius Rodrigues Veras[1,2], Pablo Holanda Cardoso[1,2], Gleyson José Pinheiro Caldeira Silva[1], Chander de Oliveira Pinheiro[3], Maria Valéria Bezerra Medina[3], Felipe Ricardo dos Santos Fernandes[1], Ingridy Marina Pierre Barbalho[1], Lyane Ramalho Cortez[1,3], João Paulo Queiroz dos Santos[1,2], Antonio Higor Freire de Morais[1,2], Gustavo Fontoura de Souza[1,2], Guilherme Medeiros Machado[4], Márcia Jacyntha Nunes Rodrigues Lucena[5], Ricardo Alexsandro de Medeiros Valentim[1]

1 Laboratory of Technological Innovation in Health (LAIS), Federal University of Rio Grande do Norte (UFRN), Natal, Rio Grande do Norte, Brazil, 2 Advanced Nucleus of Technological Innovation (NAVI), Federal Institute of Rio Grande do Norte (IFRN), Natal, Rio Grande do Norte, Brazil, 3 Secretary of Public Health of Rio Grande do Norte, Natal, Rio Grande do Norte, Brazil, 4 LyRIDS, ECE-Engineering School, Paris, France, 5 Department of Informatics and Applied Mathematics, Federal University of Rio Grande do Norte (UFRN), Natal, Rio Grande do Norte, Brazil

* tiago.barreto@lais.huol.ufrn.br

## Abstract

Bed regulation within Brazil's National Health System (SUS) plays a crucial role in managing care for patients in need of hospitalization. In Rio Grande do Norte, Brazil, the *RegulaRN Leitos Gerais* platform was the information system developed to register requests for bed regulation for COVID-19 cases. However, the platform was expanded to cover a range of diseases that require hospitalization. This study explored different machine learning models in the RegulaRN database, from October 2021 to January 2024, totaling 47,056 regulations. From the data obtained, 12 features were selected from the 24 available. After that, blank and inconclusive data were removed, as well as the outcomes that had values other than discharge and death, rendering a binary classification. Data was also correlated, balanced, and divided into training and test portions for application in machine learning models. The results showed better accuracy (87.77%) and recall (87.77%) for the XGBoost model, and higher precision (87.85%) and F1-Score (87.56%) for the Random Forest and Gradient Boosting models, respectively. As for Specificity (82.94%) and ROC-AUC (82.13%), the Multilayer Perceptron with SGD optimizer obtained the highest scores. The results evidenced which models could adequately assist medical regulators during the decision-making process for bed regulation, enabling even more effective regulation and, consequently, greater availability of beds and a decrease in waiting time for patients.

**Data Availability Statement:** The data used in this research can be accessed via the link: https://zenodo.org/records/11387710.

**Funding:** The present study was funded through the Project Regula SESAP-RN/FUNCERN, grant number 69/2021, carried out by the Laboratory of Technological Innovation in Health (LAIS) of the Federal University of Rio Grande do Norte (UFRN) in cooperation with the Secretary of Public Health of Rio Grande do Norte.

**Competing interests:** The authors have declared that no competing interests exist.

## Introduction

In Brazil, the hospital bed regulation process of the Brazilian Health System (SUS) plays a fundamental role in the management and distribution of care for patients requiring hospitalization [1, 2]. However, although the National Regulation Policy was instituted more than 15 years ago by Ordinance No. 1,559 of August 2008 [3] and consolidated by Ordinance No. 02 of September 2017 (Brazil, 2017) [4], many regions have difficulties in ensuring correct regulatory conduct.

In this context, in addition to organizational issues, the regulation system in Brazil still faces recurring problems such as the precariousness of hospital infrastructure, overcrowding in health units, an insufficient number of beds, difficulties in integration and communication between the entities involved in the regulatory process, greater transparency in processes and allocation of resources, in addition to not having efficient systems to help the regulation process [1, 5]. In Brazil, due to a non-mandatory recommendation from the Ministry of Health (MoH), the Regulation System—SISREG—is still used in many Brazilian states. This system was created in 2001 and is made available by the Brazilian Health System Informatics Department (DATASUS) [6]. Currently, this system is considered obsolete and inadequate, especially due to the lack of interoperability with the SUS technological ecosystem itself and the lack of transparency [7]. This is a legacy health information system which, although it is still used, is no longer able to play an effective role in the National Policy for the Regulation of Assistance in Access to Health Services in Brazil.

Until April 2020, the center for regulating access to health services in the state of Rio Grande do Norte did not have a platform for regulating hospital beds to systematically organize regulatory conduct within the scope of the SUS in the state. The regulatory flow control measures used were based on spreadsheets, e-mails and telephone communication, and messaging systems [8, 9].

Faced with the serious public health crisis caused by the COVID-19 pandemic, the government of the state of Rio Grande do Norte has set up technical-scientific cooperation between researchers in the field of digital health and the managers and formulators of public health policies at the State Secretariat of Public Health of RN (SESAP/RN). The aim of this technical-scientific cooperation was to formulate and implement a digital health solution that would make it possible to control and monitor the entire process of regulating hospital beds in all the state's public hospitals online, on time and transparently, totaling 24 public hospitals, with more than 900 beds available.

Based on this technical-scientific cooperation, the RegulaRN Platform for COVID-19 was developed and implemented throughout the state of Rio Grande do Norte, whose initial objective was to monitor and control access to hospital beds in wards and intensive care units (ICUs) for the disease during the pandemic [2, 8, 9]. The state of Rio Grande Norte, which is located in the northeastern region of Brazil, currently has an area of 52,797 km$^2$ and a population of approximately 3.5 million inhabitants.

After the implementation of the RegulaRN Platform, it became necessary to expand the digital health solution to the other regulatory specialties. The system is currently responsible for regulating access to beds, vascular surgery, outpatient care, exams, and consultations. In this way, the RegulaRN Platform has become a unique digital health solution for the management of health regulation services in the state of Rio Grande do Norte, an important aspect because it has centralized and integrated, through international interoperability standards, the Health Data Network (RDS) with all the other technologies in the state's public health ecosystem that are necessary for the process of regulating access to health services.

The health regulation process needs to be carried out in a rigorous, agile and transparent manner, as the incorrect conduct of a regulatory process in public health has intrinsic impacts on waiting times for access to hospital beds, as well as on hospitalization times, which can have negative impacts on the availability of hospital beds and increase the potential for existing problems [5, 10]. In this way, the inefficiency and ineffectiveness of this process can aggravate public health crisis situations, such as the COVID-19 pandemic, as it requires more rational use of health resources [8, 11–16]. Therefore, due to its complexity and the pressures that exist in all segments of the regulatory process, investment in intelligent computer systems can maximize the correct direction and assertive decision-making in healthcare systems [17–22].

Intelligent computer models have demonstrated significant potential in healthcare systems by reducing uncertainties and ambiguities in complex decision-making processes. For example, prior studies in similar healthcare contexts have shown that machine learning models can enhance hospital management by optimizing resource allocation and reducing patient waiting times [23–30]. This study aims to build on these findings to demonstrate the effectiveness of AI-based models specifically in bed regulation in Rio Grande do Norte.

In this context, the aim of this work is to analyze data from the *RegulaRN Leitos Gerais* Platform and use it to train and validate different machine learning models. Subsequently, to choose the most significant classification model capable of predicting the outcome of patients regulated by the *RegulaRN Leitos Gerais* platform with greater accuracy, precision, recall, specificity, F1 Score, and ROC-AUC. Furthermore, discuss the main impacts and potential of a digital health solution on the decision-making process of regulatory professionals.

## Materials and methods

The methodological bias of this paper consists in two main steps: exploratory data analysis and applying the data to computer models. In the evaluation process, the data was extracted, evaluated, characterized, pre-processed, and correlated. For the application stage, concerning the computational models, four phases were taken into account: 1) definition of evaluation metrics; 2) data balancing and division into training and validation groups; 3) selecting the models for classification, and 4) hyperparameters to choose the best performing model; in line with Barreto et al [2].

### Extraction, evaluation, characterization and pre-processing

This study used the database from the RegulaRN Leitos Gerais platform, a system adopted to manage the regulation flow of SUS beds in the state of Rio Grande do Norte. The database covers the period of October 2021 to January 2024, with 47.056 regulations in the two-state centers (Metropolitan and West). From this total, 1,868 regulations were removed because they were linked to newborn regulations, and these have different clinical assessment protocols when compared to adult and pediatric patients. The initial analysis therefore included 45,188 regulation requests. A more detailed descriptive analysis of the data is presented in the results section.

The initial data extraction included 24 features: a) date of request; b) occupancy type; c) case type; d) unified prioritization score (EUP); e) Sequential Organ Failure Assessment (SOFA) scale; f) type of hospital bed requested; g) admission date; h) type of input bed; i) discharge date; j) discharge bed type; k) national health card number; l) gender; m) patient's municipality; n) patient's federative unit; o) pregnant woman (yes or no); p) gestational period; q) age; r) regulator identification; s) outcome; t) requesting health unit; u) municipality of the requesting health unit; v) providing health unit; w) municipality of the providing health unit and x) ICD.

Thus, the features that were not associated with the patient's clinical condition and do not show any impact in the final result, or that relate to the locality record, such as: date of request, national health card number, patient's federative unit, patient's municipality, regulator identification, requesting health unit, municipality of the requesting health unit, and municipality of the providing health unit. In addition, features with only one possible record or insufficient information were also removed: type of occupation, type of case, pregnant woman (yes or no), and gestational age.

Consequently, only 12 characteristics were selected, namely: EUP score, SOFA scale, type of hospital bed requested, admission date, admission bed type, discharge date, discharge bed type, gender, age, outcome, providing health unit, and ICD. Using the entry date and exit date features, it was possible to create the patient's hospitalization time feature. As a result, 11 features were used in the classification process. Table 1 shows the description of all the data types extracted from RegulaRN Leitos Gerais.

**Table 1. Description of database.**

| Data description | |
|---|---|
| **Field** | **Description** |
| request date | Represents the date a bed request was registered. |
| type of occupation | Represents the type of occupation of the regulation request. |
| type of occupation | Represents the type of occupation of the regulation request. |
| case of type | Represents the results of tests for patients who were suspected of having Covid-19. The results could be: positive, negative, inconclusive, or null. |
| EUP | Represents the EUP value. The numerical scale ranges from 2 to 8 and is associated with the Charlson Comorbidity Index, Clinical Frailty Scale and simplified SOFA. |
| SOFA scale | Represents the patient's prioritization value according to the values of this scale. |
| requested bed type | Represents the type of bed selected by the regulation center for a patient. The results could be: ward and uci. |
| entry date | Represents the date the patient was allocated to the health unit (hospital). |
| entry bed type | Represents the type of bed that the patient was allocated in the health unit (hospital). The results could be: ward and uci. |
| output date | Represents the date that the patient left the bed after the outcome. |
| output bed type | Represents the type of bed the patient was in before the outcome. |
| sus card number | Represents each patient's SUS card number. Given by a 15-digit sequence. |
| sex | Represents the patient's sex. |
| patient's municipality | Indicates the patient's city of residence. |
| patient's federal unit | Is the acronym for the patient's federal unit. |
| pregnant | Represents whether the patient is pregnant or not. |
| gestational age | Describe how far along the pregnancy is, measured in weeks. |
| age | Represents patient age. |
| regulator | Represents the regulator identification responsible for regulation. |
| outcome | Represents the outcome of the patient in bed. Possible values for this field are: discharge and death. |
| requesting health unit | Represents the unit health that solicits a bed for the patient. |
| municipality of the requesting health unit | Represents the municipality of the health unit that solicits the bed. |
| provider health unit | Represents the health unit that admits and accommodates the patient in the bed. |
| municipality of the provider health unit | Represents the municipality of the health unit that receives and accomodate the patient in the bed. |
| ICD | Represents the International Classification of Diseases for bed regulation. |

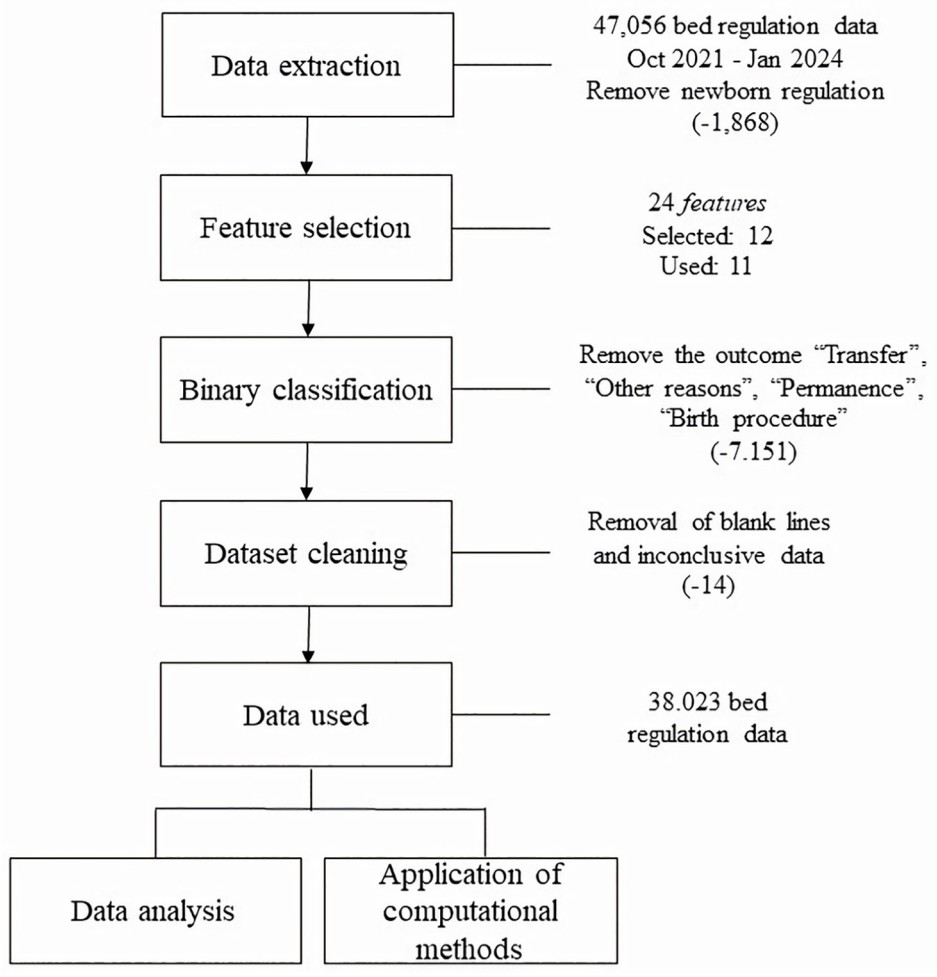

**Fig 1. Workflow defined for data processing and selection.**

After extracting the data, we evaluated the values contained in all the features and in order to guarantee the integrity of the analysis, the lines with blank data or inconclusive information were removed. In addition, the target column "outcome" contained six different values, namely: by discharge, by death, for other reasons, by stay, by delivery procedure, by transfer, etc. As these last four outcomes do not properly indicate a positive or negative closure of the regulation, as well as having a lower number of recurrences, around 7.151 regulations were removed. This maintains a binary classification (by discharge—positive, or by death—negative) for the computer models. Finally, 38.023 effective regulations were selected for application in the artificial intelligence models. Fig 1 shows the design used to process and select the data. Furthermore, in order to enable the reproducibility of this experiment, the final database used is available on the zenodo platform (https://zenodo.org/records/11387710).

## Correlation between dataset features

The first task was to perform a pairwise correlation of the features. The objective is to identify features with greater or lesser correspondence with others. As many of these are categorical data, the phik correlation model was implemented in this analysis. Phik is abble to consistently

correlate variables from several backgrounds, being categorical, ordinals and intervals a like, turning into a refinement of Pearson [31] hypothesis test.

## Definition of evaluation metrics

The overall aim of the study is to classify hospital bed regulation data to predict a patient's positive or negative outcome. Furthermore, it is important to investigate the models' performance in situations where predictions are wrong, either due to a high number of false positives or false negatives. Thus, it is necessary to include not only accuracy, but also precision, recall, specificity, F1-Score, and ROC-AUC in a similar way to those found in the works of Iwendi et al [32], Aljameel et al [33] and Endo et al [34].

The accuracy consists in the set of data with correct predictions (true positive and true negative) divided by the sum of all predictions made by the model (true positive, true negative, false positive, false negative) (Eq 1):

$$Accuracy = (TP + TN)/(TP + FP + FN + TN) \qquad (1)$$

Precision consists of dividing the true positive rate by the sum of the true positive and false positive rates (Eq 2).

$$Precision = TP/(TP + FP) \qquad (2)$$

Recall involves the rate of true positives divided by the rate of true positives plus false negatives (Eq 3).

$$Recall = TP/(TP + FN) \qquad (3)$$

Specificity refers to the prediction of true negatives divided by the sum of true negatives and false positives (Eq 4).

$$Specificity = TN/(TN + FP) \qquad (4)$$

The F1-score is the harmonic mean between the precision and recall. The formula involves the product of precision and recall divided by the sum of these metrics, multiplied by 2 (Eq 5).

$$F1Score = 2 * (Precision * Recall)/(Precision + Recall) \qquad (5)$$

ROC-AUC can be obtained by recall divided by the complementary value of specificity (Eq 6).

$$ROCAUC = Recall/(1 - Specificity) \qquad (6)$$

## Data balancing and splitting into training and validation data

The RegulaRN Leitos Gerais Platform database refers to real-world bed regulation data, in this sense, there is an unbalanced distribution of data when classified by outcome, 82.6% are discharges and 17.4% deaths. The use of an unbalanced database biases the machine learning classifiers, making the algorithms able to identify patterns from the predominant class much better than patterns from the minority class. To mitigate this problem, one of the most common techniques is SMOTE (Synthetic Minority Over-sampling), which works by increasing the number of data points in the minority class [35]. The SMOTE algorithm first identifies the minority class, then in the feature vector space identifies the k nearest neighbors of that class (k is usually equal to 5). Finally, a new instance of the minority class is generated by randomly

selecting values in the vector space between an instance of the minority class and the nearest neighbors identified. This process is repeated until the database is completely balanced.

In addition, as for the division of training and validation data, the same segmentation was used as in other studies applying machine learning techniques that use a large volume of data [36–38]. Therefore, 80% of the data was directed to training and the others 20% for validation.

## Definition of models for data classification

The selection of classification models was based on their proven ability to handle large volumes of imbalanced healthcare data [37, 39–41]. Decision tree was selected because it is one of the classic models that handles high volumes of data well and has wide application in problems in the health area [42]. Random Forest, on the other hand, was selected for its ability to manage complex decision trees and its resistance to overfitting, particularly in high-dimensional datasets [43]. Gradient Boosting and Adaboost due to their adaptability and efficient ability to capture non-linear relationships [44, 45]. XGBoost, for example, has been shown to perform well in healthcare settings due to its gradient boosting framework, which effectively handles missing data and provides robust performance on tabular datasets [46] and Multi-Layer Perceptron (MLP) has an architecture capable of modeling non-linear relationships and performing gradient learning, adjusting weights efficiently for larger volumes of data [47]. For the MLP models, two different paths were taken, the Stochastic Gradient Descent (SGD) was selected due to its performance and Adam because of its consistency in treating gradient explosion and fading problems [48]. These models were chosen for their complementarity in addressing the specific challenges of bed regulation data in this study.

## Hyperparameters to define the best model

After the model selection, it was necessary to define the best combination of hyperparameters to enhance the evaluation metrics of each model. Thus, this section presents which hyperparameters were adopted and which methods were elaborated in the training and validation steps. It is worth mentioning that all computational model development in this research used python's sckit-learn library [49].

For each selected model, hyperparameters were set aiming to boost the performance metrics. In this regard, the following hyperparameters were selected for Decision Tree: criterion, which measures the quality of node splitting; max depth of tree, which determines the maximum depth of the tree; min samples leaf, which represents the minimum number of samples needed in a leaf; and max features, which considers the maximum number of features analyzed to perform a split. For the Random Forest and Gradient Boosting models, the criterion, max depth of the tree and max features were also used, including the number estimators, which considers the number of trees in the forest. In the Adaboost model, the parameters number estimators, learning rate and algorithm were chosen. The learning rate refers to the learning weight at each iteration, while the algorithm relates to how the model can speed up the convergence of the classifier with the least possible error. For XGBoost: learning rate, number estimators, max depth and colsample by tree. This last hyperparameter is associated with the randomly selected fraction of resources that will be used to train each tree. Finally, for the MLP Adam and MLP SGD models, the following were used: hidden layer size, which represents the number of layers in the model; activation, which represents the model's activation function and batch size, which represents the size of the minibatches that will be used to help the optimizers.

The grid GridSearchCV functionality, which allows all possible combinations of hyperparameters to be iterated, was applied during the training to find which parameters showed better

**Table 2. Selection of hyperparameters and values for each model.**

| Models | Hyperparameter | Range and best values |
|---|---|---|
| Decision Tree | criterion | gini or entropy; |
| | max depth of the tree | [10, 50, 100]; |
| | min samples leaf | range [1, 2, 3, 4]; |
| | max features | [sqrt, log2]. |
| Random Forest | criterion | gini or entropy; |
| | max depth of the tree | [10, 50, 100]; |
| | number estimators | [100, 200, 400]; |
| | max features | [sqrt, log2]. |
| Gradient Boosting | criterion | friedman_mse or squared_error; |
| | max depth | [10, 50, 100]; |
| | number estimators | [10, 50, 100]; |
| | max features | [sqrt, log2]. |
| Adaboost | learning rate | [0.1, 0.5, 1.0] |
| | number estimators | [100, 200, 400] |
| | algorithm | [samme, samme.r] |
| XGBoost | learning rate | [0.1, 0.5, 1.0] |
| | number estimators | [100, 200, 400] |
| | max depth | [10, 50, 100] |
| | colsample by tree | [0.1, 0.5, 1.0] |
| MLP SGD | hidden_layer_sizes | [5, 25, 70] |
| | activation | tanh or relu |
| | batch_size | [16, 32, 64] |
| MLP ADAM | hidden_layer_sizes | [5, 25, 70] |
| | activation | tanh or relu |
| | batch_size | [16, 32, 64] |

results in the evaluated metrics [50, 51]. A proportional division of the training and test data was also carried out randomly using the cross validation attribute with a value of 10-folds in the GridSearchCV functionality, as a way of enhancing the model's learning. In addition, the models were trained five times, similar to that developed by Ahsan et al [52], in order to determine the best set of hyperparameters more accurately. The details of the hyperparameters used and the respective values chosen for each model are shown in Table 2.

## Results

### General data analysis

Considering the data profile from *RegulaRN Leitos Gerais*, between October 2021 and January 2024, it was possible to identify that most hospitalizations involve male adults, young people and children, in hospital beds and with lower EUP score and SOFA scale. The details of the values extracted from the database are presented in Table 3, classifying each of the characteristics based on their respective outcome.

In addition, the database contains outcomes by each provider hospital (41), to address which health units had the highest number of requests and their respective outcomes, given that his feature showed a high correlation with several other dataset variables. Each hospital has a different treatment specialty, and thus some receive requests of greater complexity and mortality than others, culminating in different proportions of discharges and deaths. Finally,

**Table 3. Data profile from the RegulaRN Leitos Gerais.**

| Features | | Values | Outcomes |
|---|---|---|---|
| Age | ≥ 60 | 18.641 | Discharge: 13.549 |
| | | | Death: 5.092 |
| | < 60 | 19.382 | Discharge: 17.860 |
| | | | Death: 1.522 |
| Sex | Masculine | 20.235 | Discharge: 16.941 |
| | | | Death: 3.384 |
| | Feminine | 17.698 | Discharge: 14.468 |
| | | | Death: 3.230 |
| EUP Score | 2 | 17.572 | Discharge:16.668 |
| | | | Death: 904 |
| | 3 | 7.279 | Discharge: 6.176 |
| | | | Death: 1.103 |
| | 4 | 5.468 | Discharge:4.113 |
| | | | Death: 1.355 |
| | 5 | 5.485 | Discharge: 3.543 |
| | | | Death:1.942 |
| | 6 | 1.765 | Discharge: 770 |
| | | | Death: 995 |
| | 7 | 367 | Discharge: 120 |
| | | | Death: 247 |
| | 8 | 87 | Discharge:19 |
| | | | Death: 68 |
| SOFA | 1 | 29.699 | Discharge: 26.070 |
| | | | Death: 3.629 |
| | 2 | 6.816 | Discharge: 4.652 |
| | | | Death: 2.164 |
| | 3 | 1.179 | Discharge: 598 |
| | | | Death: 581 |
| | 4 | 329 | Discharge: 89 |
| | | | Death: 240 |
| requested bed type | Ward | 23.863 | Discharge: 21.878 |
| | | | Death: 1.985 |
| | ICU | 14.160 | Discharge: 9.531 |
| | | | Death: 4.629 |
| entry bed type | Ward | 23.858 | Discharge: 21.851 |
| | | | Death: 2.007 |
| | ICU | 14.165 | Discharge: 9.558 |
| | | | Death:4.607 |
| output bed type | Ward | 28.137 | Discharge: 26.233 |
| | | | Death: 1.904 |
| | ICU | 9.886 | Discharge: 5.176 |
| | | | Death: 4.710 |

(*Continued*)

**Table 3.** (Continued)

| Features | | Values | Outcomes |
|---|---|---|---|
| Provider health unit | Health unit 1 | 1.234 | Discharge: 1.061 |
| | | | Death: 173 |
| | Health unit 2 | 794 | Discharge: 554 |
| | | | Death: 240 |
| | Health unit 3 | 299 | Discharge: 213 |
| | | | Death:86 |
| | Health unit 4 | 225 | Discharge: 154 |
| | | | Death: 71 |
| | Health unit 5 | 367 | Discharge: 227 |
| | | | Death:140 |
| | Health unit 6 | 2.104 | Discharge: 1.722 |
| | | | Death: 382 |
| | Health unit 7 | 2.865 | Discharge: 2.341 |
| | | | Death: 524 |
| | Health unit 8 | 2.331 | Discharge: 1.971 |
| | | | Death: 360 |
| | Health unit 9 | 112 | Discharge: 104 |
| | | | Death: 8 |
| | Health unit 10 | 8 | Discharge: 7 |
| | | | Death: 1 |
| | Health unit 11 | 2.138 | Discharge: 2.094 |
| | | | Death: 44 |
| | Health unit 12 | 703 | Discharge: 625 |
| | | | Death: 78 |
| | Health unit 13 | 101 | Discharge: 56 |
| | | | Death: 45 |
| | Health unit 14 | 11 | Discharge: 11 |
| | | | Death: 0 |
| | Health unit 15 | 422 | Discharge: 225 |
| | | | Death:197 |
| | Health unit 16 | 94 | Discharge: 86 |
| | | | Death: 8 |
| | Health unit 17 | 5 | Discharge: 3 |
| | | | Death: 2 |
| | Health unit 18 | 385 | Discharge:233 |
| | | | Death:152 |
| | Health unit 19 | 703 | Discharge:703 |
| | | | Death:0 |
| | Health unit 20 | 253 | Discharge:211 |
| | | | Death:42 |
| | Health unit 21 | 2.825 | Discharge:2.646 |
| | | | Death:179 |
| | Health unit 22 | 1.000 | Discharge:778 |
| | | | Death:222 |

(*Continued*)

**Table 3.** (Continued)

| Features | | Values | Outcomes |
|---|---|---|---|
| | Health unit 23 | 586 | Discharge:373 |
| | | | Death:213 |
| | Health unit 24 | 739 | Discharge:608 |
| | | | Death:131 |
| | Health unit 25 | 737 | Discharge:631 |
| | | | Death:106 |
| | Health unit 26 | 34 | Discharge:23 |
| | | | Death:11 |
| | Health unit 27 | 396 | Discharge:173 |
| | | | Death:223 |
| | Health unit 28 | 5.192 | Discharge:4.320 |
| | | | Death:872 |
| | Health unit 29 | 740 | Discharge:662 |
| | | | Death:78 |
| | Health unit 30 | 1.000 | Discharge:850 |
| | | | Death:150 |
| | Health unit 31 | 598 | Discharge:438 |
| | | | Death:160 |
| | Health unit 32 | 2.183 | Discharge:1.590 |
| | | | Death:593 |
| | Health unit 33 | 395 | Discharge:240 |
| | | | Death:155 |
| | Health unit 34 | 443 | Discharge:233 |
| | | | Death:210 |
| | Health unit 35 | 193 | Discharge:193 |
| | | | Death:0 |
| | Health unit 36 | 4.181 | Discharge:3.774 |
| | | | Death:407 |
| | Health unit 37 | 422 | Discharge:422 |
| | | | Death:0 |
| | Health unit 38 | 12 | Discharge:12 |
| | | | Death:0 |
| | Health unit 39 | 522 | Discharge:467 |
| | | | Death:55 |
| | Health unit 40 | 161 | Discharge:119 |
| | | | Death:42 |
| | Health unit 41 | 510 | Discharge:256 |
| | | | Death:254 |
| Length of stay | < 7 | 16.693 | Discharge: 13.658 |
| | | | Death:3.035 |
| | $7 \leq LoS \leq 14$ | 11.499 | Discharge: 9.824 |
| | | | Death: 1.675 |
| | >14 | 9.831 | Discharge: 7.927 |
| | | | Death: 1.904 |
| Outcomes | Discharge | 31.409 | |
| | Death | 6.614 | |

the data includes around 2055 different diseases classified by the International Classification of Diseases (ICD-10), which were also examined for recurrence.

As for the statistical profile, the average age is 53.38 years, with a standard deviation of 26.82 years and a median of 59 years. The average hospitalization time was 12.96 days, with a standard deviation of 17.67 days and a median of 7 days. The mean EUP score was 3.15 and the median was 3. The mean SOFA scale was 1.2 and the median 1.

Regarding the ICD, Table 4 shows the ten most recurrent ICDs, followed by the municipalities with the highest incidence and the hospitals that treat the most. The state capital, the city of Natal, has the highest number of inhabitants and has the highest incidence of ICDs 6 and 10. In contrast, Mossoró, the second largest municipality in terms of inhabitants, has a higher incidence in four of the ten. The noteworthy point is that there is no significant number of requests for these diseases among Parnamirim, São Gonçalo do Amarante, and Macaíba municipalities, which are the 3rd, 4th and 5th most populous municipalities.

SOFA and EUP are two tools used to evaluate the hospital's bed priority for each patient, considering that EUP revolves around SOFA, The Charlson Comorbidity Index (CCI) and the

**Table 4. Distribution of the most frequent ICDs by municipality and hospital.**

| Code | Name | Frequency | Municipality with the highest incidence | Hospitals that treat the most |
|---|---|---|---|---|
| J18.9 | Unspecified Pneumonia | 4284 | Natal: 1437 | Provider Hospital 28: 504 |
| | | | Mossoró: 763 | Provider Hospital 6: 344 |
| | | | Santo Antônio: 278 | Provider Hospital 11: 333 |
| I21.9 | Unspecified Acute Myocardial Infarction | 1738 | Mossoró 781 | Provider Hospital 36: 910 |
| | | | Natal: 150 | Provider Hospital 6: 158 |
| | | | Currais Novos:97 | Provider Hospital 7: 119 |
| I64 | Stroke, Not Specified as Hemorrhagic or Ischemic | 1540 | Mossoró: 1010 | Provider Hospital 28: 969 |
| | | | Caicó: 150 | Provider Hospital 32: 164 |
| | | | Natal: 148 | Provider Hospital 6: 73 |
| N39.0 | Urinary Tract Infection of Unspecified Localization | 965 | Natal: 408 | Provider Hospital 21: 125 |
| | | | Mossoró: 83 | Provider Hospital 41: 102 |
| | | | Parnamirim: 79 | Provider Hospital 30: 67 |
| I50.0 | Congestive Heart Failure | 905 | Natal: 243 | Provider Hospital 6: 134 |
| | | | Mossoró: 196 | Provider Hospital 28: 73 |
| | | | Currais Novos: 97 | Provider Hospital 21: 68 |
| I20.0 | Unstable Angina | 677 | Mossoró: 531 | Provider Hospital 36: 529 |
| | | | Natal: 53 | Provider Hospital 6: 30 |
| | | | Currais Novos: 8 | Provider Hospital 21: 13 |
| F20.8 | Other Schizophrenias | 587 | Natal: 292 | Provider Hospital 21: 293 |
| | | | Parnamirim: 50 | Provider Hospital 7: 272 |
| | | | Ceará-Mirim: 34 | Provider Hospital 6: 9 |
| A46 | Erysipelas | 546 | Natal: 180 | Provider Hospital 21: 85 |
| | | | Mossoró: 70 | Provider Hospital 32: 64 |
| | | | Caicó: 62 | Provider Hospital 39: 33 |
| A41.9 | Unspecified Septicemia | 542 | Natal: 151 | Provider Hospital 32: 81 |
| | | | Mossoró: 115 | Provider Hospital 28: 72 |
| | | | Caicó: 67 | Provider Hospital 6: 38 |
| I25.2 | Old Myocardial Infarction | 510 | Mossoró: 373 | Provider Hospital 36: 374 |
| | | | Natal: 31 | Provider Hospital 7: 19 |
| | | | João Câmara: 19 | Provider Hospital 25: 17 |

**Table 5. Distribution of the ICDs with the highest number of deaths.**

| Code | Name | Frequency | Deaths/Total incidence |
|------|------|-----------|------------------------|
| J18.9 | Unspecified Pneumonia | 1052 | 24.5% |
| A41.9 | Unspecified Septicemia | 279 | 50.5% |
| I50.0 | Congestive Heart Failure | 247 | 27.3% |
| J18.0 | Unspecified Bronchopneumonia | 110 | 24.8% |
| I21.9 | Unspecified Acute Myocardial Infarction | 94 | 12.6% |

Clinical Frailty Scale (CFS). According to the data, it is possible to identify that EUP, has a more normalized aggregation to the outcomes classification. Meanwhile, SOFA = 1, was responsible for characterizing 78% of the data, a similar percentage is presented in the sum of requests with EUP 2 (46.2%), 3 (19.1%), and 4 (14.3%). In other words, while SOFA indicates that 78% of referrals had the same degree of priority, EUP structures the same percentage into three different categories. Given the health sector's peculiarities, the EUP is an indicator that minimizes the generalization of different clinical conditions.

Another important point to evaluate is the ICD that most frequently resulted in death. Naturally, each ICD has its own intrinsic lethality level, meaning that some diseases kill more than others. However, it is necessary to analyze the frequency of certain occurrences and whether the incidence is local and already expected by public health institutions. Hence, with the data in hand, public health authorities can evaluate and orchestrate future intervention proposals. As shown in Table 5, Unspecified Pneumonia (J18.9) was the disease with the highest frequency (see Table 4) and resulted in the most deaths. Around 24.5% of the patients classified with this disease died, resulting in 15.9% of the total number of deaths. However, Unspecified Septicemia is one of the most lethal diseases and is responsible for the death of 50.5% of patients diagnosed with this disease.

Regarding the data correlation, shown in Fig 2, Phik's correlation revealed that the features that relate most closely to the outcome are the output bed type, requested bed type, entry bed type, SOFA scale, ICD, age, EUP score, and provider unit. Length of stay and gender did not present any relevant correlation for this topic.

## Machine learning model results

Table 2 shows the selection of hyperparameters that indicated the best results for the selected models. The Decision Tree model obtained the best criterion results when the entropy node division strategy was selected, max depth of the tree with a value of 50, min samples leaf with a value of 1 and max feature, square root. Random Forest obtained the best results with entropy (criterion), 50 (max depth of tree), 400 (number estimators) and sqrt (max features). For Gradient Boosting, squared error (criterion), 10 (max depth of tree), 50 (number estimators) and sqrt (max features). In the Adabost model, the best results were: 1.0 (learning rate), 400 (number estimators), and samme.r (algorithm). In XGBoost: 0.1 (learning rate), 200 (number estimators), 50 (max depth) and 1.0 (colsample by tree). The MLP models used the same hyperparameters in SGD and ADAM (hidden layer sizes, activation, batch size) which resulted in the same values: 70, relu and 32.

As for the results obtained by the selected metrics, XGBoost scored highest in accuracy (87.77%) and recall (87.77%). On the other hand, the Random Forest model (87.85%) was the most accurate, i.e. being the model that best classifies the positive outcome. As for the F1-Score value, the Gradient Boosting model had the highest value (87.56%). As for specificity, a parameter that assesses the classification performance of the negative outcome, it can be seen

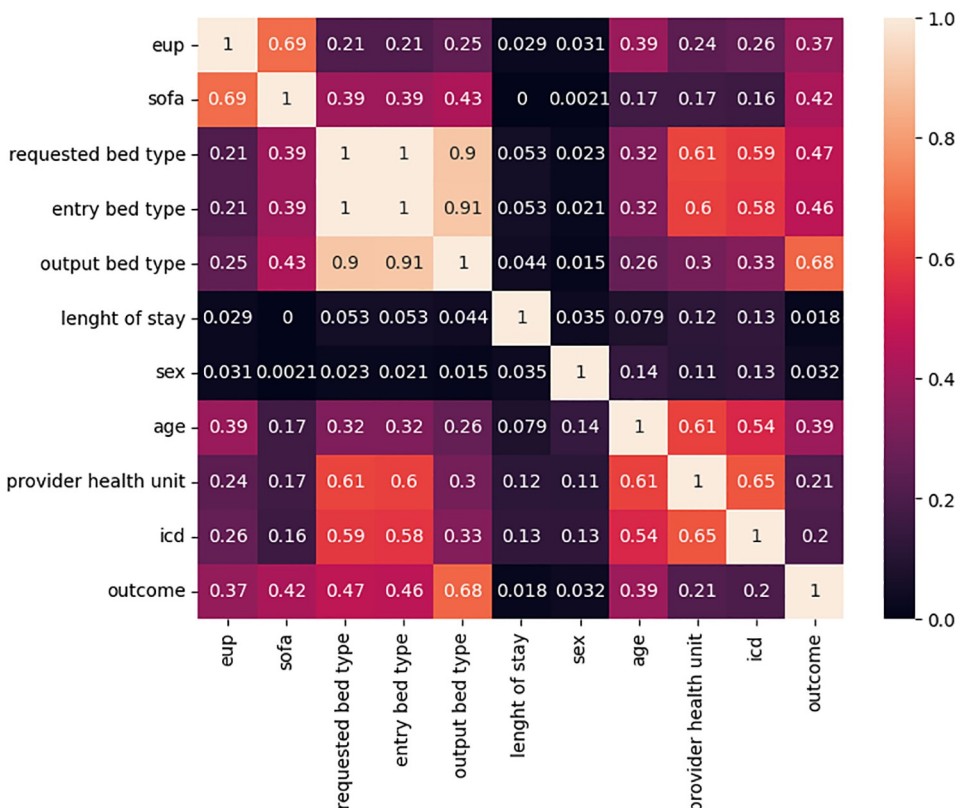

**Fig 2. Presentation of the Phik correlation for RegulaRN Leitos Gerais data.**

that the multilayer perceptron models outperform the others. The highest score was obtained by the SGD (82.94%). Table 6 presents the performance metrics for each machine learning model, including accuracy, precision, recall, F1-Score, and specificity. Notably, XGBoost outperformed the other models in accuracy and recall, making it a robust choice for predicting patient outcomes in bed regulation. However, the high specificity observed in the MLP models indicates that these models may be more suitable when the goal is to minimize false positives, particularly in critical care cases. For a better comparison of the performance of the models used, Fig 3 presents the values of each metric per computational model.

Based on the results of the models, we performed a chi-square statistical validation to analyze whether the behavior of the models has statistical significance. For this, a contingency

**Table 6. Metrics obtained by the computer models.**

| Models | Accuracy | Precision | Recall | F1-Score | Specificity |
|---|---|---|---|---|---|
| Decision Tree | 82.97(+0.13) | 84.26(+0.19) | 82.96(+0.13) | 83.51(+0.14) | 64.36(+0.42) |
| Random Forest | 87.20(+0.01) | **87.85(+0.03)** | 87.20(+0.01) | 87.47(+0.01) | 72.98(+0.17) |
| Gradient Boosting | 87.14(+0.05) | 88.21(+0.03) | 87.14(+0.05) | **87.56(+0.03)** | 75.47(+0.24) |
| Adaboost | 86.69(+0.05) | 87.76(+0.05) | 86.69(+0.05) | 87.12(+0.05) | 74.25(+0.06) |
| XGBoost | **87.77(+0.07)** | 87.46(0.04) | **87.77(+0.07)** | 87.60(+0.10) | 66.96(+0.04) |
| MLP SGD | 83.36(+0.17) | 88.10(+0.07) | 83.36(+0.17) | 84.73(0.13) | **82.94(+0.50** |
| MLP ADAM | 82.88(+0.76) | 87.87(+0.13) | 82.88(+0.76) | 84.33(+0.61) | 82.58(+0.62) |

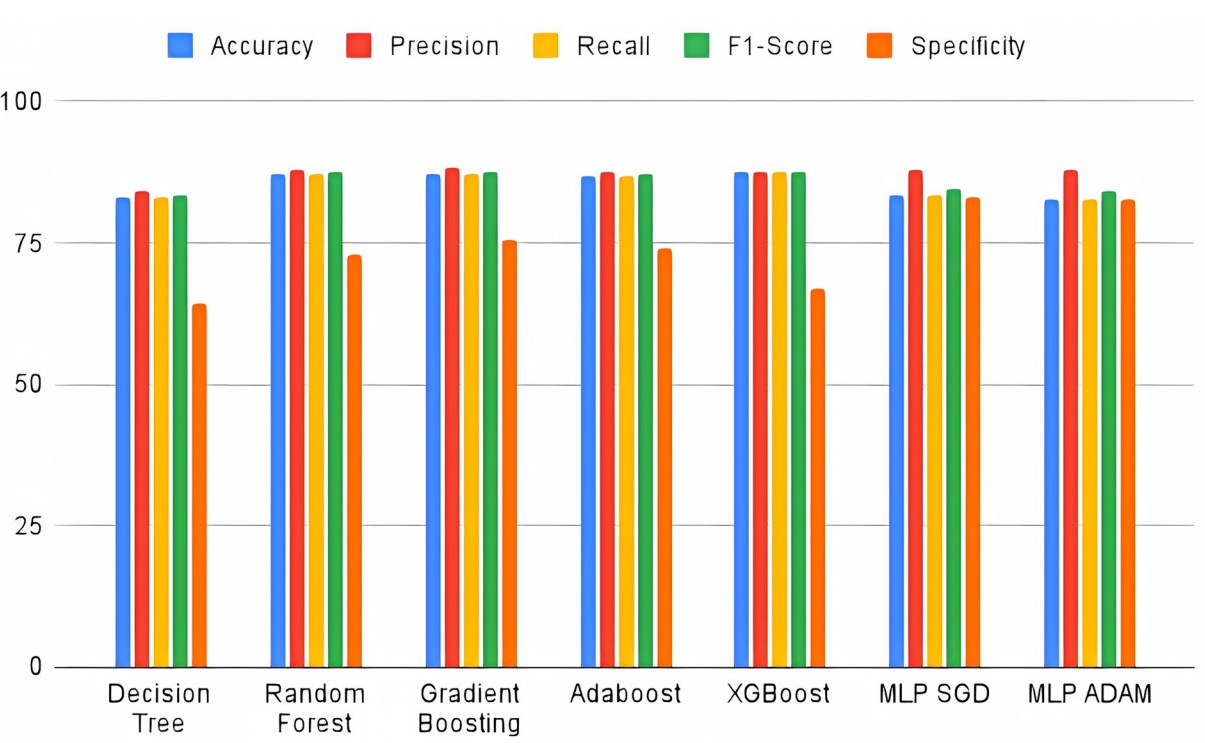

**Fig 3. Comparison of the performance of the models used.**

table was created with the distribution of real and predicted values of all models and for all cases a p value < 0.01 was obtained.

As for the features that were important for training the models, the most relevant features for classifying the Decision Tree were *bed type*, *age*, *provider health unit* and *icd*. The non-relevant elements were *requested bed type*, *entry bed type* and *SOFA*. In the Random Forest model, *output bed type*, *age*, *EUP*, *provider health unit* and *icd* scored the highest, while *sex* and *SOFA* were the least relevant characteristics. For the Gradient Boosting classifier *output bed type*, *age*, *EUP* and *provider health unit* were the most relevant, while *sex*, *requested bed type*, *entry bed type* were the lowest scorers. Adaboost considered the best characteristics to be *length of stay*, *provider health unit*, *EUP* and *age*, while the least relevant were *sex*, *requested bed type* and *entry bed type*. XGBoost considered *output bed type*, *EUP* and *entry bed type* as the most important characteristics and *sex*, *SOFA* and *requested bed type* as the least important. For the models that used MLP, Adam considered *output bed type*, *requested bed type*, *provider health unit* and *age* to be the most relevant, while SGD considered *output bed type*, *provider health unit*, *age* and *ICD* to be the most significant. The least important features were *entry bed type* and *sex* for Adam; and *sex* and *requested bed type* for SGD. Figs 4 and 5 show the important features of the machine learning models.

Compared to Phik's correlation, except for Adabost, all the other classifiers included *output bed type* as the most important feature in the classification process, which corroborates Phik's correlation (*output bed* being the feature with the highest correlation with the outcome) and the weaker correlation, *sex* was identified as the least relevant feature for Random Forest, Grandient Boosting, Adaboost, XGBoost and SGD, while *length of stay*, which is another feature that has been shown to have a low correlation with the outcome, was not identified as worse in any of the classifiers, however, for Adaboost this feature was the most significant.

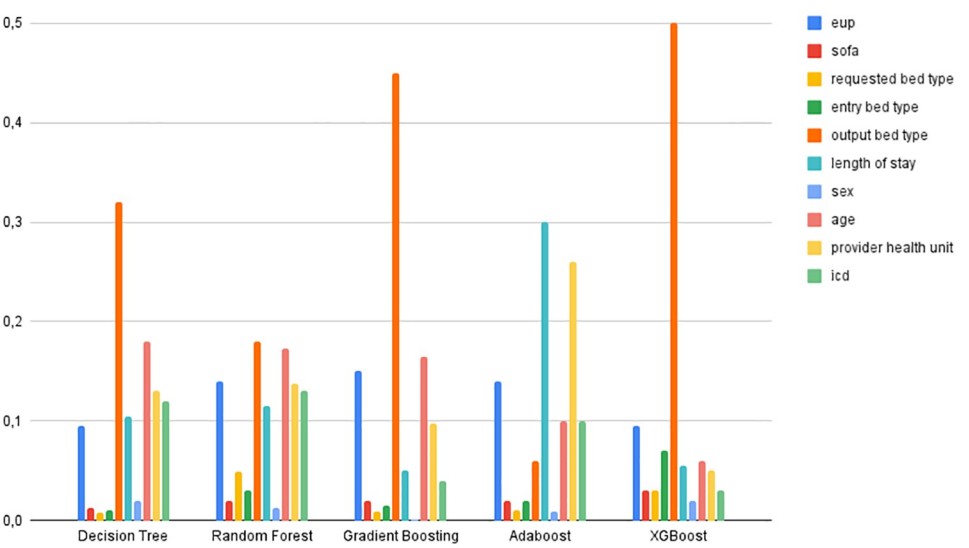

**Fig 4. Feature importance of the machine learning models.**

The ROC curve (receiver operating characteristic curve) helps to visualize the performance of classifiers to select an appropriate operating point or decision threshold [53]. The discriminative capacity is usually quantified by the area under the AUC curve when considering the prediction of a binary event. It relates the variation in the rate of true positives and false positives predicted by the models, with results on a scale of 0 to 1. Although there is no definitive consensus in the literature, most studies using this tool consider an AUC between 0.7 and 0.8 to be good and acceptable, and between 0.8 and 0.9 to be very good [54, 55].

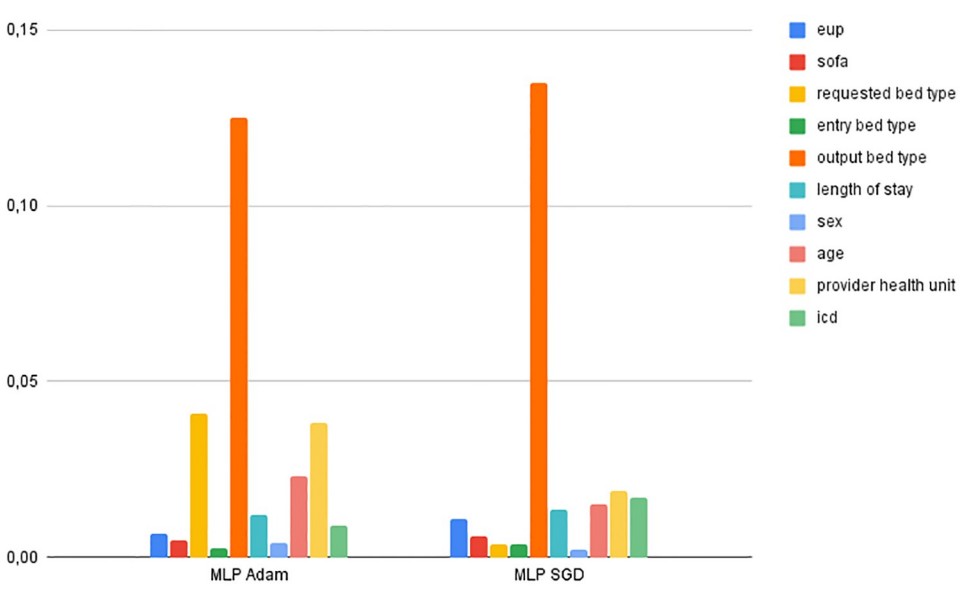

**Fig 5. Feature importance of MLP models.**

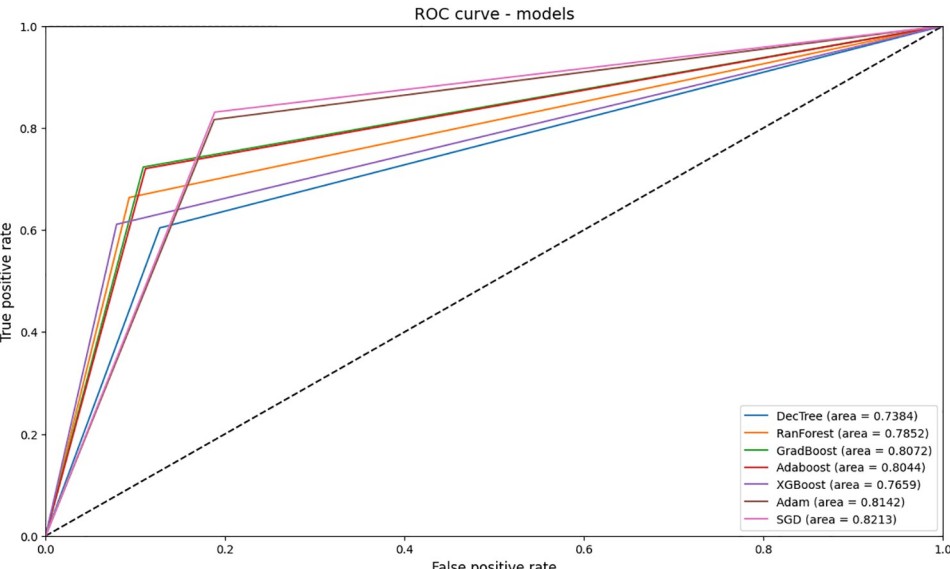

**Fig 6. ROC curve and AUC value of all models.**

Thus, the Decision Tree (AUC = 0.738), XGBoost (AUC = 0.766) and Random Forest (AUC = 0.785) models performed well, while the Adaboost (AUC = 0.804), Adam (AUC = 0.814) and SGD (AUC = 0.821) models performed better, falling into the very good category. Fig 6 shows the results obtained.

## Discussion

The use of artificial intelligence and computational methods to solve and predict problems in the health field has been going on for some years now, and although there is a considerable range of solutions in various segments, from predicting diseases by diagnosing medical images [56–60] to the classification of important markers for the prediction of cardiological [25, 61], and ophthalmological diseases [62, 63] or the analysis of data to predict early-stage cancer [64–66]; as well as robotic mechanisms for surgery, for example [67–70]. There are still some sectors that have not been explored or that have made negligible contributions [57, 71–73].

According to Yu, Beam and Kohane [57], the association of artificial intelligence will be able to contribute even more effectively to clinical practices and health management. In this way, healthcare professionals will be able to reduce the time spent on repetitive tasks in order to explore better treatments and clinical solutions aimed at patient care, something that machines cannot do and which require more humanized treatment. According to Valentim et al [8], the use of digital health solutions based on artificial intelligence are already considered relevant tools by healthcare managers, as they help to make decision-making more timely, effective and based on robust scientific evidence.

In the process of regulating hospital beds, the use of artificial intelligence helps to reduce medical subjectivity in the face of the repetitive process of countless daily regulations, tasks that can often become a tiring activity throughout the day. This certainly contributes to minimizing errors in the indication of hospital beds, especially when it comes to public health, since the daily volume of care is extremely high, as is the case in the state of Rio Grande Norte in Brazil, which has a population of approximately 3.5 million inhabitants. This could result in

better resolutions for patients, as well as better equity in access to the resources available in the public health system. All of this will lead to a more timely hospitalization process for patients, and consequently to better performance in terms of hospital bed turnover—better average occupancy time for hospital beds across the entire public health network [2]. In general, the use of machine learning tools can optimize the care process, increasing efficacy, efficiency and effectiveness, which induces better resilience of the health system, especially in times of crisis, as was the case during COVID-19 [8, 74, 75].

At the management level, adopting AI-driven systems for bed regulation could lead to significant improvements in resource allocation, reducing patient wait times and optimizing bed occupancy rates. However, implementing these systems at scale presents challenges, such as ensuring adequate training for health professionals and integrating AI tools with existing hospital infrastructure. Addressing these challenges will be critical for maximizing the potential benefits of AI in the public healthcare system [5, 8].

In this study, machine learning techniques were used in different tree and ensemble models, as well as artificial neural network models on hospital bed regulation data, and the aim was to classify the outcome of patients regardless of their ICD, to help the regulating doctor and reduce subjectivity during the hospital bed regulation process.

As for the results of the computer models, XGBoost showed the best accuracy (87.77%) and recall (87.77%) values, i.e. of all the models used, it classifies the data better in general, regardless of the outcome (discharge or death), as well as, given the positive outcome, the proportion that was correctly classified. As for the accuracy indicator, which identifies which proportion of positive outcomes was correct, Random Forest performed best (87.05%). As for the F1-Score, Gradient Boosting has a better harmonic mean between precision and recall, i.e. it has a better balance in the metrics that assess the positive outcome. Regarding specificity, a metric that assesses the classification of the negative outcome, the neural network models showed the best results when compared to the tree and ensemble models, achieving scores of 82.58% (ADAM) and 82.94% (SGD). For the ROC-AUC, the SGD and ADAM models also performed better, because, as they had a more balanced classification of positive and negative outcomes, the ROC-AUC value was in the range of 82.13% and 81.42%, respectively.

Considering these results, the models used in this experiment are not only able to predict which patients are more likely to be discharged or die, but also allow us to understand which samples are being better classified concerning the outcome and the best type of hospital bed according to the clinical conditions of each patient. And so, the main metric analyzed should not only be accuracy; the other metrics that point to a positive outcome (precision, recall and ROC-AUC) should also be maximized [2]. Furthermore, it also has a positive impact on the pace of work of the regulatory professional, given that in situations of high demand and overload of requests, the assertiveness of the regulatory process can be compromised, and so the models contribute to better regulatory conduct [76, 77].

## Conclusion

This study used the regulation database of the *RegulaRN Leitos Gerais* platform between 2021 and 2024 in machine learning models to predict the outcome of discharge and death in different diseases that require hospitalization. The results of this article show that there is no single model that obtains the best accuracy, precision, recall, F1-Score, specificity, and ROC-AUC metrics. Thus, depending on the objectives of the regulatory professionals, it should be observed which model can provide the best result based on the desired metric, i.e. for example, if the regulator's objective is to observe the best classifications for the positive outcome, it

should use XGBoost and Random Forest; If the objective is to evaluate the best classification for the negative outcome, the multilayer perceptron models should be evaluated.

It should be noted that artificial intelligence computer models enhance the activities carried out in the healthcare and management sectors. Research in this area should therefore be increasingly explored in order to minimize the precariousness and weaknesses that exist in the different health segments. In this way, this research also aims to make a positive contribution to the health system such as the SUS, which aims to guarantee universal and comprehensive access to health with equity.

A significant limitation of this study is the incomplete dataset, particularly the absence of detailed information such as pregnancy status and gestational age. This missing information could introduce bias in model predictions, particularly for patient subgroups with different clinical needs. Future work should focus on improving data collection protocols to ensure that such critical variables are recorded, allowing for more nuanced and accurate model predictions across diverse patient groups. During the evaluation of the database, some gaps were found in the data, which is why it was not included for training the models. However, for some diseases, knowing whether the patient is pregnant or not and the appropriate length of pregnancy are essential. In addition, this study considered the same evaluation of hospital outcomes in different diseases with different morbidity scales. Furthermore, another limitation of this work was the non-inclusion of other models widely used in academic literature, such as k-Nearest Neighbors (kNN) and Support Vector Machines (SVM) [78–80], as they were not included within the initial scope of this research. However, it is considered that for future work the scope of computational models can be expanded and these models included. Furthermore, still addressing future work, creating a new feature that can categorize diseases by morbidity could contribute to a more appropriate classification of the models. Furthermore, trying to identify which treatment protocols were used to treat certain diseases can also be a relevant indicator for classifying models.

## Acknowledgments

We would like to thank the Public Health Secretariat of Rio Grande do Norte (SESAP/RN), the Health Technological Innovation Laboratory (LAIS) of the Federal University of Rio Grande do Norte (UFRN), the Advanced Innovation Center (NAVI) of the Federal Institute do Rio Grande do Norte (IFRN), to LyRIDS, ECE-Engineering School and the Department of Informatics and Applied Mathematics for the support necessary for the development of this research.

## Author Contributions

**Conceptualization:** Tiago de Oliveira Barreto, Fernando Lucas de Oliveira Farias, Felipe Ricardo dos Santos Fernandes, Antonio Higor Freire de Morais, Ricardo Alexsandro de Medeiros Valentim.

**Data curation:** Tiago de Oliveira Barreto, Fernando Lucas de Oliveira Farias, Nicolas Vinícius Rodrigues Veras, Pablo Holanda Cardoso, Gustavo Fontoura de Souza.

**Formal analysis:** Tiago de Oliveira Barreto.

**Funding acquisition:** Chander de Oliveira Pinheiro, Lyane Ramalho Cortez.

**Investigation:** Fernando Lucas de Oliveira Farias, Gleyson José Pinheiro Caldeira Silva, Felipe Ricardo dos Santos Fernandes, Ingridy Marina Pierre Barbalho, Márcia Jacyntha Nunes Rodrigues Lucena, Ricardo Alexsandro de Medeiros Valentim.

**Methodology:** Tiago de Oliveira Barreto, Nicolas Vinícius Rodrigues Veras, Pablo Holanda Cardoso, Gleyson José Pinheiro Caldeira Silva, Chander de Oliveira Pinheiro, Maria Valéria Bezerra Medina, Felipe Ricardo dos Santos Fernandes, Ingridy Marina Pierre Barbalho, Lyane Ramalho Cortez, João Paulo Queiroz dos Santos, Antonio Higor Freire de Morais, Gustavo Fontoura de Souza, Guilherme Medeiros Machado, Márcia Jacyntha Nunes Rodrigues Lucena, Ricardo Alexsandro de Medeiros Valentim.

**Project administration:** João Paulo Queiroz dos Santos.

**Resources:** Tiago de Oliveira Barreto, Nicolas Vinícius Rodrigues Veras, Pablo Holanda Cardoso, Antonio Higor Freire de Morais.

**Validation:** Lyane Ramalho Cortez, Guilherme Medeiros Machado.

**Visualization:** Gleyson José Pinheiro Caldeira Silva, Maria Valéria Bezerra Medina, Ingridy Marina Pierre Barbalho.

**Writing – original draft:** Tiago de Oliveira Barreto, Fernando Lucas de Oliveira Farias, Nicolas Vinícius Rodrigues Veras, Pablo Holanda Cardoso, Gleyson José Pinheiro Caldeira Silva, Ricardo Alexsandro de Medeiros Valentim.

**Writing – review & editing:** Tiago de Oliveira Barreto, Fernando Lucas de Oliveira Farias, Nicolas Vinícius Rodrigues Veras, Chander de Oliveira Pinheiro, Maria Valéria Bezerra Medina, Felipe Ricardo dos Santos Fernandes, Ingridy Marina Pierre Barbalho, Lyane Ramalho Cortez, João Paulo Queiroz dos Santos, Antonio Higor Freire de Morais, Gustavo Fontoura de Souza, Guilherme Medeiros Machado, Márcia Jacyntha Nunes Rodrigues Lucena, Ricardo Alexsandro de Medeiros Valentim.

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
