## [Decision Letter · Decision Letter 0]

24 Sep 2024

PONE-D-24-27498Artificial Intelligence Applied to Bed Regulation in Rio Grande do Norte: Data Analysis and Application of Machine Learning on the “RegulaRN Leitos Gerais” PlatformPLOS ONE

Dear Dr. Barreto,

Thank you for submitting your manuscript to PLOS ONE. After careful consideration, we feel that it has merit but does not fully meet PLOS ONE’s publication criteria as it currently stands. Therefore, we invite you to submit a revised version of the manuscript that addresses the points raised during the review process.

I hope this message finds you well. I have reviewed your study on improving technology for bed regulation within a universal health system and would like to offer some suggestions to enhance its international relevance.

To strengthen your findings, I recommend providing a more detailed explanation of the current limitations of the national system used for bed regulation. Additionally, contextualizing these regulations within the broader health system would provide valuable insights.

As per the reviewers' feedback, please update the discussion with current literature and enhance the comprehensibility of the methods and results section. Including a flowchart and improving the graphics and tables will also aid in conveying your findings more effectively.

I look forward to seeing the revised manuscript.

We look forward to receiving your revised manuscript.

Kind regards,

Luísa da Matta Machado Fernandes, DrPH

Academic Editor

PLOS ONE

Journal Requirements:

 [copy in funding statement]. 

Reviewers' comments:

Reviewer's Responses to Questions

**Comments to the Author**

1. Is the manuscript technically sound, and do the data support the conclusions?

Reviewer #1: Partly

Reviewer #2: Partly

Reviewer #3: Yes

2. Has the statistical analysis been performed appropriately and rigorously? 

Reviewer #1: No

Reviewer #2: No

Reviewer #3: Yes

3. Have the authors made all data underlying the findings in their manuscript fully available?

Reviewer #1: Yes

Reviewer #2: No

Reviewer #3: Yes

4. Is the manuscript presented in an intelligible fashion and written in standard English?

Reviewer #1: Yes

Reviewer #2: Yes

Reviewer #3: Yes

5. Review Comments to the Author

Reviewer #1: This study effectively applies machine learning to the RegulaRN Leitos Gerais platform to optimize hospital bed regulation in Rio Grande do Norte. Analyzing data from October 2021 to January 2024, it shows strong performance from models like XGBoost, Random Forest, and Gradient Boosting in accuracy, precision, and recall. However, the study would benefit from incorporating additional methods such as k-Nearest Neighbors (kNN) and Support Vector Machines (SVM), as well as referencing recent literature (e.g., doi.org/10.1109/ACCESS.2024.3392729 and doi.org/10.1016/j.patcog.2023.109641) to ensure a comprehensive evaluation and alignment with current advancements in healthcare machine learning.

Reviewer #2: • Suggested Improvement: The introduction clearly outlines the general problem of bed regulation but could be more detailed in discussing the relevance of artificial intelligence in this specific context by comparing it with similar studies.

o Lines 60-61: "Intelligent computer models can help reduce the impact of uncertainties and ambiguities in the regulatory process and improve decision-making support."

o Replace with: "Intelligent computer models have demonstrated significant potential in healthcare systems by reducing uncertainties and ambiguities in complex decision-making processes. For example, prior studies in similar healthcare contexts have shown that machine learning models can enhance hospital management by optimizing resource allocation and reducing patient waiting times (reference). This study aims to build on these findings to demonstrate the effectiveness of AI-based models specifically in bed regulation in Rio Grande do Norte."

More detailed discussion on model limitations The article mentions limitations but could provide a more robust discussion of how these limitations affect the results and suggest strategies to mitigate them.

• Line 398: "The study’s limitations include the fact that the health professionals did not provide more precise information on some of the data that could have been better analyzed in the models, such as whether the patient was pregnant."

• Replace with: "A significant limitation of this study is the incomplete dataset, particularly the absence of detailed information such as pregnancy status and gestational age. This missing information could introduce bias in model predictions, particularly for patient subgroups with different clinical needs. Future work should focus on improving data collection protocols to ensure that such critical variables are recorded, allowing for more nuanced and accurate model predictions across diverse patient groups."

More detailed explanation of the choice of machine learning models The choice of machine learning models is explained, but a more in-depth justification of why certain models (like XGBoost and Random Forest) were selected would be helpful.

• Line 165: "The definition of models for data classification involved algorithms that, according to the literature, perform well with high volumes of data."

• Replace with: "The selection of classification models was based on their proven ability to handle large volumes of imbalanced healthcare data. XGBoost, for example, has been shown to perform well in healthcare settings due to its gradient boosting framework, which effectively handles missing data and provides robust performance on tabular datasets (Chen & Guestrin, 2016). Random Forest, on the other hand, was selected for its ability to manage complex decision trees and its resistance to overfitting, particularly in high-dimensional datasets (Breiman, 2001). These models were chosen for their complementarity in addressing the specific challenges of bed regulation data in this study."

Better organization of tables and figures The presentation of tables and figures can be improved with more descriptive captions and the inclusion of an analysis immediately after presenting each figure/table to facilitate the interpretation of the results.

• Line 276: "Table 6 shows all the values obtained."

• Replace with: "Table 6 presents the performance metrics for each machine learning model, including accuracy, precision, recall, F1-Score, and specificity. Notably, XGBoost outperformed the other models in accuracy and recall, making it a robust choice for predicting patient outcomes in bed regulation. However, the high specificity observed in the MLP models indicates that these models may be more suitable when the goal is to minimize false positives, particularly in critical care cases."

More in-depth discussion of the practical impact of the results The discussion could delve deeper into the practical impact of adopting AI in the healthcare system and the challenges of implementing it on a large scale.

• Line 349: "At the management level, in response to a more efficient regulatory system, financial and human resources can be distributed in a way that is more coherent with the needs of the different health sectors."

• Replace with: "At the management level, adopting AI-driven systems for bed regulation could lead to significant improvements in resource allocation, reducing patient wait times and optimizing bed occupancy rates. However, implementing these systems at scale presents challenges, such as ensuring adequate training for health professionals and integrating AI tools with existing hospital infrastructure. Addressing these challenges will be critical for maximizing the potential benefits of AI in the public healthcare system."

I noticed the absence of graphs and visual figures that could significantly improve the clarity and understanding of the article, especially concerning the methodology and results. The inclusion of graphical elements such as flowcharts and result charts would be a valuable contribution to make the methodological process more accessible and the data easier to interpret.

Firstly, presenting a detailed methodology flowchart would be extremely helpful in guiding readers through all the stages described in the article. This would clarify the process from data extraction and processing to machine learning model selection, allowing for a clearer understanding of the methodological sequence.

Additionally, to facilitate the understanding of the results, it would be interesting to include graphs comparing the performance metrics of the different machine learning models. Bar graphs, for example, could visually show the accuracy, recall, specificity, and F1-Score of the tested models. The inclusion of ROC-AUC curves would also help visually present each model’s discriminative capabilities, making the information easier to interpret for readers.

Finally, the article mentions correlation analysis between variables, but including a correlation matrix, such as a heatmap, could visually highlight the most significant relationships between the variables. This type of graphical visualization would make data interpretation more immediate and intuitive for the reader.

These visual additions, such as flowcharts and result charts, would not only enhance the manuscript’s clarity but also help more effectively convey the complexities involved in data analysis and the results obtained.

Reviewer #3: Texto apresenta clareza de dados, necessidade de pequenos ajustes apontados no manuscrito enviado. A publicação duplicada, intencional ou não, pode prejudicar a credibilidade da pesquisa e comprometer os direitos de propriedade intelectual de ambos os periódicos.

6. PLOS authors have the option to publish the peer review history of their article (what does this mean?). If published, this will include your full peer review and any attached files.

Reviewer #1: No

Reviewer #2: No

Reviewer #3: **Yes: **Danielle Torres dos Santos Lopes

---

## [Author Response · Author response to Decision Letter 0]

11 Oct 2024

A response to each suggestion was attached as a response document to the reviewers.

---

## [Editor Report · Decision Letter 1]

26 Nov 2024

Artificial Intelligence Applied to Bed Regulation in Rio Grande do Norte: Data Analysis and Application of Machine Learning on the “RegulaRN Leitos Gerais” Platform

PONE-D-24-27498R1

Dear Dr. Barreto,

We’re pleased to inform you that your manuscript has been judged scientifically suitable for publication and will be formally accepted for publication once it meets all outstanding technical requirements.

Kind regards,

Luísa da Matta Machado Fernandes, DrPH

Academic Editor

PLOS ONE
---

## [Editor Report · Acceptance letter]

28 Nov 2024

PONE-D-24-27498R1 

PLOS ONE

Dear Dr. Barreto, 

I'm pleased to inform you that your manuscript has been deemed suitable for publication in PLOS ONE. Congratulations! Your manuscript is now being handed over to our production team.

Kind regards, 

on behalf of

Dr. Luísa da Matta Machado Fernandes 

Academic Editor

PLOS ONE